# Sequence, Chromatin and Evolution of Satellite DNA

**DOI:** 10.3390/ijms22094309

**Published:** 2021-04-21

**Authors:** Jitendra Thakur, Jenika Packiaraj, Steven Henikoff

**Affiliations:** 1Department of Biology, Emory University, Atlanta, GA 30322, USA; jenika.s.packiaraj@emory.edu; 2Basic Sciences Division, Fred Hutchinson Cancer Research Center, Seattle, WA 98109, USA; steveh@fredhutch.org; 3Fred Hutchinson Cancer Research Center, Howard Hughes Medical Institute, Seattle, WA 98109, USA

**Keywords:** repetitive DNA, heterochromatin, centromeres, telomeres, H3K9me3, CENP-A, non-B-form DNA

## Abstract

Satellite DNA consists of abundant tandem repeats that play important roles in cellular processes, including chromosome segregation, genome organization and chromosome end protection. Most satellite DNA repeat units are either of nucleosomal length or 5–10 bp long and occupy centromeric, pericentromeric or telomeric regions. Due to high repetitiveness, satellite DNA sequences have largely been absent from genome assemblies. Although few conserved satellite-specific sequence motifs have been identified, DNA curvature, dyad symmetries and inverted repeats are features of various satellite DNAs in several organisms. Satellite DNA sequences are either embedded in highly compact gene-poor heterochromatin or specialized chromatin that is distinct from euchromatin. Nevertheless, some satellite DNAs are transcribed into non-coding RNAs that may play important roles in satellite DNA function. Intriguingly, satellite DNAs are among the most rapidly evolving genomic elements, such that a large fraction is species-specific in most organisms. Here we describe the different classes of satellite DNA sequences, their satellite-specific chromatin features, and how these features may contribute to satellite DNA biology and evolution. We also discuss how the evolution of functional satellite DNA classes may contribute to speciation in plants and animals.

## 1. Introduction

Eukaryotic DNA sequences are present either as single copies in the genome, or as repeats, which can be either interspersed throughout the genome or locally repeated in tandem. Interspersed repeats mostly comprise transposable elements and their relics, which are scattered throughout the genome, whereas tandem repeats are found mostly at or around centromeres and telomeres. These distinct classes of repetitive DNA were characterized almost two decades before the introduction of Sanger and Maxam–Gilbert DNA sequencing based on analysis of reannealing kinetics [1]. When DNA is denatured and allowed to reanneal, single-copy DNA anneals slowly because the DNA strand complementary to any given strand is present at low concentration, whereas the concentration of the complementary strand doubles when it is present in two copies, and so on. Unlike bacterial genomic DNA, which anneals as a single component, DNA from multicellular eukaryotes, such as mammals, anneals as three components: The slowly annealing single-copy DNA typically accounts for as much as half of the total, with the remainder being middle-repetitive dispersed repeats and highly repetitive tandem repeats. The highly repetitive fraction was also identified by buoyant density gradient centrifugation. A high-concentration aqueous solution of cesium chloride (CsCl) centrifuged at high speed will form a gradient, and where DNA bands in a CsCl gradient at equilibrium is determined by its base composition. The term “satellite DNA” was coined to describe DNA with a base composition different enough from most other DNA in the genome that it forms a satellite band above (AT-rich DNA) or below (GC-rich DNA) the main band containing the bulk of the genome [1]. Subsequently, restriction enzyme digestion and electrophoresis of the DNA fragments generated on agarose gels followed by Southern blot hybridization using DNA probes were used to clone and characterize repetitive DNA [2,3,4,5]. With the advancement of genomics with next-generation sequencing technologies and bioinformatics tools for repeat analysis (e.g., Tandem Repeat Finder [6], RepeatExplorer [7], TAREAN [8] etc.), the cataloging of the repetitive regions has gained momentum, which together with genetic and cell biological approaches has revealed that many satellite families are involved in distinct cellular functions.

## 2. Satellite DNA Sequences

Satellite DNAs are classified into microsatellites, minisatellites, satellites and macrosatellites based on the monomeric repeat length. Microsatellites, most relevant to medicine, also called simple sequence repeat (SSR) or short tandem repeat (STR), are small (2–6 bp in size) tandem repeats. Microsatellite copy numbers are characteristically variable within a population. The human genome contains ~3% microsatellites [9,10]. Telomeric regions are occupied by microsatellites that can be repeated multiple times, eventually forming the bulk of the telomeric region up to 15 kb on human chromosomes [11]. Due to their polymorphic nature microsatellites are used in gene discovery by linkage analysis for identification purposes in paternity testing or forensic DNA testing [12,13,14]. Minisatellite monomeric repeat units are larger (~15 bp) but fewer in number than microsatellites. Minisatellite arrays exhibit a mean length of 0.5–30 kb and are found in euchromatic regions of the genome of vertebrates, fungi and plants. Several thousand chromosomal loci are enriched with minisatellites in the human genome [15,16]. Minisatellites are highly variable in array size and classified as variable number of tandem repeats (VNTRs). In the human genome, VNTRs have been used for DNA fingerprinting to type individuals [17]. Each organism has a unique arrangement pattern, the only exception being multiple individuals from a single zygote (e.g., identical twins) [18]. Satellites consist of tandemly repeated arrays that are present at centromeres, pericentromeric regions and subtelomeric regions. Most satellite arrays are characterized by either simple repeats or complex repeats that involve increase in repeat unit length and complexity by merging shorter repeat motifs [19,20,21,22,23,24]. Macrosatellite monomer repeat units are much larger and range up to a few kilobases in length [25]. Examples of human macrosatellites are D4Z4, a 3.3 kb macrosatellite located at the subtelomeric regions of chromosomes 4q35 and 10q26, DXZ4, a 3 kb CpG-rich macrosatellite is present in 12–100 tandem copies on chromosome Xq23 and NBL2, a 1.4 kb macrosatellite repeat mainly found on the short arm of acrocentric chromosomes 13, 14, 15 and 21 [25,26]. Satellite DNA exhibits a high degree of polymorphism due to variations in the number of repeat units caused by mutations involving several mechanisms [27].This hypervariability among satellite DNA repeats of related and unrelated organisms makes them excellent markers for mapping, characterizing the genomes, genotype-phenotype correlation, marker-assisted selection of the crop plants, molecular ecology and diversity-related studies [14,15,16,28].

### 2.1. Functional Satellite Classes

#### 2.1.1. Centromeric Satellites

Centromeres are the sites of spindle binding and are located within the primary constrictions, the “pinched-waist” of chromosomes. Spindles bound to centromeres pull replicated chromatids/homologous chromosomes apart to mediate faithful chromosome segregation during mitosis and meiosis. Centromere function in chromosome segregation is conserved across eukaryotes, but centromeric satellites are the fastest evolving genomic elements and closely related species exhibit dramatic variations in their centromeric DNA sequences [29,30]. Centromeric sequences in the budding yeast *Saccharomyces cerevisiae* are non-repetitive and are bound by centromeric proteins in a sequence-dependent manner [31]. However, centromeres of most other eukaryotes are occupied by arrays of satellite repeats that span up to several megabases [2,32,33]. The best studied centromeric satellites are those found in primates, mice, plants and insects (Table 1).

##### Primate Centromeric Satellites

Centromeres of primates consist of α-satellites that may constitute as much as 10% of the total number of repeats in primate genomes [46]. α-satellites are the most abundant satellites in the human genome representing half of the total human satellite DNA [10,47]. Individual monomers in the human α-satellite pool are up to 60% divergent and have been classified by Alexandrov into five suprachromosomal groups or families, based on the organization of repeat monomers [19,48]. Suprachromosomal families 1-2 (SF1-2) are tandem arrays of two alternating α-satellite units. Two α-satellite units of a dimer can differ up to ~30%. These dimeric units were independently identified as the most abundant functional centromere associated α-satellites based on their binding to centromere marker proteins in cell lines from individuals belonging to five different populations [49]. Dimeric arrays are present on multiple chromosomes and are highly enriched for centromeric proteins, therefore forming the majority of “functional” α-satellites [49]. One of the two α-satellite units of a dimer contain the CENP-B box, the binding site for Centromere Protein B (CENP-B) [50,51]. The SF3 family contains a higher-order repeat (HOR) organization, which results from the simultaneous amplification and homogenization of a block of more than two α-satellite monomers [19,48,52]. As a result, individual α-satellite repeat units within a HOR are typically more divergent (~70% identical) than adjacent HOR blocks (up to 97–100% sequence identity) [53,54]. Unlike SF1-2 dimers, SF3 α-satellite HORs are mostly chromosome-specific. Centromeric arrays of chromosomes 1, 11, 17 and X contain SF3 HORs in which the repeating unit is a block of five monomers (pentameric) [48]. The HOR with highest number of repeating units are present on the centromere of chromosome Y (34-mer HOR) [55]. A recent study has shown that even on HORs, the most prominent basic pattern is the dimeric unit with the presence of a CENP-B box at every alternative monomer [56]. The remaining α satellites are present as monomers and do not bind to enough centromeric proteins to form functional centromeres. The HOR organization had been recently found to exist in hominoids as well as the Old and new world monkeys [57,58,59,60,61].

On rare occasions, when a human native satellite centromere is inactivated, centromere function is acquired by an ectopic non-satellite locus called neocentromere [21,62,63]. More than 100 cases of human neocentromeres have been studied and, in most cases, they were found to occur in regions that are gene deserts [64,65]. Neocentromeres can also be experimentally induced by deleting native centromeres [64,65]. In some experimentally tractable organisms, neocentromeres form primarily at locations adjacent to the native centromeres or in the subtelomeric regions [66,67,68,69].

##### Mouse Centromeric Satellites

Mouse chromosomes are telocentric, with centromeric satellite arrays located at or near one end of the chromosome. Despite sharing ~70% sequence identity in their genes, mice and humans carry completely different satellite repeats. *Mus musculus* centromeres consist of tandem arrays of 120-bp minor satellite (MiSat), which contains the CENP-B box and comprise 1–2% of the genome [33,34]. MiSats are more homogeneous within a *Mus* species as compared to α-satellites of primate species [24,33,34,70]. Additional centromeric satellites are the ~150-bp MS3 and the ~300-bp MS4, both of which cytologically co-localize with MiSats at *M. musculus* centromeres [35,71]. The association of MS3 and MS4 with functional centromeric markers remains to be elucidated, but both of these satellites contain CENP-B-like boxes indicating that they might act as functional centromeric satellites [35]. *M. spretus*, which separated from *M. musculus* ~1-2 MYA, has retained a small fraction of MiSats, which are distributed throughout the entire centromeric and pericentric domain, unlike in *M. musculus* where MiSats are exclusively within the centromeric regions [70,72,73]. *M. caroli*, separated from *M. musculus* ~6 MYA, lacks minor satellites and has evolved different families of satellites with monomer lengths of 60 and 79 bp, which localize to the telocentric ends [72,73]. The *M. caroli*-specific 79-bp satellite monomer includes a distant variant of the CENP-B box, which contains all nine bases necessary for its interaction with Cenp-B, and is capable of binding to *M. musculus* Cenp-B in an interspecific cell line [50].

##### Plant Centromere Satellites

Similar to primate and mouse centromeric satellites, the centromeric repeat units in most flowering plants are AT-rich in composition and approximately nucleosomal in length (Table 1). Centromeric regions of several plant species are colonized by retroelements that belong to the Ty3/gypsy endogenous retrovirus family [30,74,75,76,77,78,79,80]. The integrases of plant centromeric retroelements contain a chromodomain, which is thought to ensure correct integration of the retrotransposon to the centromeric region and are collectively referred to as putative centromeric targeting domains [74,75,81]. Maize (*Zea mays*) satellite repeat, CentC, is a 156-bp monomeric repeat unit and is highly interspersed with centromeric retrotransposons of maize (CRM) [36]. Interspersed CentC/CRM arrays range from ~300–3000 kbp [82]. *Z. mays* contain a nonessential selfish B chromosome, which is highly heterochromatic and maintained in populations through a system of non-Mendelian inheritance. The B chromosome contains a neocentromere, the 370-kb Pmel fragment, which is important for maize B chromosome transmission and shares high sequence similarity to repeats in knobs of maize standard chromosomes [37,83,84]. Rice (*Oryza sativa*) centromeres are dominated by a 155-bp rice-specific CentO satellite repeat unit and a centromere-specific retrotransposon, CRR [39,85]. CentO is present in ~6000 copies in the rice genome that share 85–90% sequence identity [39,85]. The holocentric plant *Luzula nivea* and nine other *Luzula* species contain a 178 bp repeat unit, LCS1, which is similar to rice CentO and is potentially centromeric in origin [40,86,87]. *Arabidopsis thaliana* centromeres consist of tandem arrays of a 178-bp pAL1 repeat unit spanning ~1–3 Mb and comprising ~3% of the genome [32,41]. Other members of the *Arabidopsis* genus contain pAL1-like tandem repeats that are ~60–80% identical [88].

##### Drosophila Centromeric Satellites

Among nearly 1 million known insect species, satellite DNA has been studied for Diptera, Hymenoptera, Bombyx and Tribolium [30], and functional centromeric satellites have been functionally characterized in *D. melanogaster* and its sister species, *D. simulans* (Table 1). *Drosophila* centromeric satellites differ from satellites that dominate centromeres of most other complex eukaryotes in that repeat units are not of nucleosomal length, but rather are very short, mostly 5–10 bp. In early studies, a region sufficient for centromere function centromeres in *D. melanogaster* was mapped on an X-derived minichromosome, Dp1187, to a 420-kb locus containing the AAGAG and AATAT satellites interspersed with “islands” of complex sequences [89,90]. *D. melanogaster* centromeric satellite repeats are mostly tandem arrays of short 5- to 12-bp sequences, often following the pattern RRNRN, where R is a purine and N is any nucleotide [91]. Two recent studies have characterized *D. melanogaster* and *D. simulans* functional centromeres based on their binding to centromeric protein marker CENP-A [44,45]. Our group found that *D. melanogaster* centromeres are largely occupied by the Prodsat 10-mer AATAACATAG, AATAT, AATAG, and AAGAT repeats and have diverged in their sequences and numbers in *D. simulans*. *D. simulans* centromeric regions are enriched in a simple 10-mer AATAGAATTG repeat and two complex sequence families simcent1 and simcent2 [45]. Both the simcent1 and simcent2 families, contain a local tandem duplication of a core sequence of ~76 bp (simcent1) or 44 bp (simcent2) within the longer complex sequence [45]. Highly abundant Prodsat comprises roughly 5 ± 2% of the *D. melanogaster* genome but is absent in other species. Chang et al. found that all *D. melanogaster* centromeres are occupied by a non–long terminal repeat (non-LTR) retroelement in the Jockey family, G2/Jockey-3, which is also present at the centromeres of *D. simulans* [44]. It is possible that G2/Jockey-3 has evolved to target centromeres, regions that lack meiotic recombination and so may serve as “safe havens” for parasitic elements. This finding is reminiscent of the recent colonization of maize centromeres by the CR2 retrotransposon [92]. It will be interesting to determine whether the presence of G2/Jockey-3 has acquired any functional role at *Drosophila* centromeres.

#### 2.1.2. Pericentromeric Satellites

Pericentric regions flank centromeres and are bound by cohesin, which prevents premature separation of sister chromatids during chromosome segregation. Pericentric satellites are the most abundant satellites in their genomes and in mouse constitute ~10% of the total. In interphase nuclei, pericentromeric satellites from multiple chromosomes are clustered into chromocenters in several eukaryotes including mouse, flowering plants and *Drosophila* [93,94,95,96]. In *D. melanogaster*, defective chromocenter formation results in micronuclei formation due to budding from the interphase nucleus, DNA damage and cell death [96].

##### Mammalian Pericentric Satellites

Unlike human centromeric regions, which are occupied by a single satellite family (α-satellite), pericentric regions are occupied by several different satellite families (Table 2). Pericentric regions of human chromosomes 3, 4, 9, 13, 14, 15, 21, and 22 contain satellites I, II, and III, which are composed of a mixture of simple short repeated sequences satellites 1 (17 bp, 25 bp), 2 (10–80 bp) and 3 (5 bp, 10 bp), respectively [22,97,98,99,100,101,102,103]. A subset of GC-rich β-satellites (Sau3A DNA family) with 68-bp monomer repeat units are also present at the pericentric regions of multiple chromosomes (chromosome 9; acrocentrics—chromosomes 13, 14, 15, 21, and 22) and the secondary constriction of chromosome 1 [23,104,105]. Pericentromeric regions of human chromosomes 8, X, and Y contain tandem arrays of 220-bp GC-rich γ-satellite usually forming 10- to 200-kb clusters flanked by α-satellite DNA [104,106]. Two acrocentric chromosomes can fuse to form a metacentric chromosome through Robertsonian (Rb) translocation in humans [107,108,109,110,111,112]. Human Rb translocation breakpoints predominantly span the pericentric regions of acrocentric chromosomes 13, 14, and 21 and involve pericentric satellites [108,109,113,114]. The majority of the Rb translocations retain satellite III DNA [109]. Several properties of satellite DNA, such as extensive sequence identity between chromosomes and large uninterrupted arrays, make them ideal for non-homologous recombination and Rb translocations. Similarly, mouse telocentric ends that span pericentric regions are hotspots for Rb translocations in wild populations of *M. musculus* located in Western Europe and North Africa [109,115]. Mouse pericentric regions are occupied by major satellites (MaSats,), the TRPC-21A family repeats and TLC (TeLoCentric) repeats [24,34]. MaSat is the most abundant mouse satellite comprising ~10% of the genome and is present at pericentric regions of all chromosomes. The MaSat repeat unit is 234 bp long, is composed of four different 58-60 bp simple repeats, and exhibits monomer variability of up to 30% [24,34,116]. While the MaSat satellite is confined to the pericentric regions in *M. musculus*, it is also present in the chromosomal arms in *M spretus* and *M. caroli* [20]. The TRPC-21A family repeat, which has a basic 21-bp repeat unit, resembles human pericentric satellites (Sat I, II and III) and is more GC-rich than MiSat and MaSat [24]. The TLC satellite has a 146-bp repeat unit and is located between MiSat and telomeres [24]. The subgenus *Mus* includes 14 species separated into three geographic clades: Southeast Asian, Indian and Palearctic (which includes *M. musculus*). MaSat and MiSat are only found in the Indian and Palearctic clades, whereas the TLC satellite is present in all three clades but absent in some species (*M. famulus*, *M. cypriacus*, and *M. m. musculus*) within each clade [73]. This suggests that TLC must have appeared in the ancestor of the *Mus* subgenus and was then lost independently in some species within each clade. MiSat and MaSat might have arisen and amplified after the divergence of the Southeast Asian clade. This may also suggest that TLC satellites gave rise to MiSat and MaSat sequences before the divergence of the Indian and Palearctic groups, as TLC sequences have higher sequence similarity with MiSat and MaSat than they have between each other [73].

##### Plant Pericentric Satellites

The majority of flowering plant species in which pericentric satellites have been characterized carry species-specific multiple pericentric satellites similar to those of insects [117,118,119,129]. *Arabidopsis* pericentric regions are occupied by multiple satellite families including 180-bp repeat family pAL1/pAtMr, Athila retrotransposons, 500-bp and 160-bp tandem repeat families [117,118,119]. The 500-bp repeats co-localize with centromeres, while the 160-bp repeats localize adjacent to the centromeres. Interestingly, the 500-bp repeat unit is related to pAL1 and proposed to have originated by duplication of one-half of an ancestral pAL1 repeat followed by insertion of a foreign element [117]. *Arabidopsis* pericentric regions are also occupied by four dispersed repeat units, which average 1–2 kb in length. These dispersed repeats are present with copy numbers of 20–300 and are estimated to span up to 1.5 Mb on each side of the *Arabidopsis* centromeres [117]. Pericentric satellites of potato *Solanum bulbocastanum* carry a unique 5.9-kb tandem repeat 2D8 repeat, which is highly homologous to the intergenic spacer (IGS) of the 18S.25S ribosomal RNA genes. Each 2D8 unit is arranged as a dimer of ~3 kb monomers showing some divergence from one another and differing in AT-content. Sequence, structure and copy number of the 2D8 repeat is highly variable throughout the Solanum genus, suggesting that it is evolutionarily dynamic [120].

##### Insect Pericentric Satellites

In insects, pericentric satellites have been well-studied in many species (Table 2). Similar to centromeric regions, pericentric regions of *D. melanogaster* contain several different satellite repeats—the 1.688 gm/cm^3^ buoyant density family of related satellite repeats (353-bp and 356-bp repeats on chromosome 3, a 260-bp repeat on chromosome 2 and a 359-bp repeat on chromosome X), Prod satellite [(AATAACATAG)_n_], Responder (Rsp) satellite (a dimer of two 120-bp monomers that share 84% identity), AAGAT, AATAG and AAGAG repeats [121,122,123]. The 359-bp satellite-containing array is located immediately adjacent to the rDNA locus on the X chromosome and is suggested to play a role in regulating the expression of the rDNA genes [130]. Rsp satellite located on chromosome 2 is highly variable, ranging from ~10 to over 3000 repeats per array among individuals and can be deleterious in specific genetic backgrounds [131]. Although most Rsp satellites are located in the pericentric regions, a small number of Rsp satellites are also present in euchromatic loci throughout the genome [132]. When the homologous chromosome 2 carries a Segregation Distorter (SD) allele, Rsp satellite causes spermatids bearing it to degenerate causing loss of half of sperms, which results in high transmission frequencies of the SD-carrying chromosome [133]. Retention of Rsp satellite in the *Drosophila* genome despite its harmful effects has been suggested due to its requirement for the organismal fitness as deletion of Rsp satellite array is associated with a reduction in the fitness [134].

The pericentric satellite repeats of *Chrysolina* (leaf beetles) species are highly species-specific (Table 2). *C. carnifex* pericentromeric species-specific complex satellite monomers are organized into three types of repeats: monomers (211-bp) and higher-order repeats in the form of dimers (477-bp) or trimers (633-bp) [126]. *Palorus subdepressus* (tenebrionid beetles) pericentric regions of all 20 chromosomes contain a highly abundant 72-bp satellite that comprises 20% of the genome [124]. The 72-bp *P. subdepressus* pericentric repeat is composed of two copies of T2A5T octanucleotide alternating with 22-bp units of an inverted repeat. Interestingly, phylogenetic clustering of *P. subdepressus* pericentric satellite sequence variants separates them into two clades organized in alternating dimeric 144-bp repeat units [124].

The red flour beetle *Tribolium castaneum* satellite repeats TCAST1 and TCAST2 comprise >35% of the genome and are present at both centromeric as well as pericentromeric regions of all 20 chromosomes [127,135,136]. Interestingly, evolutionarily young variants of TCAST2 have been detected in the natural populations of *T. castaneum*. These populations carry three distinct TCAST2 subfamilies, which differ in monomer size, genome organization, and subfamily specific mutations [128]. Subfamilies Tcast2a (359-bp repeat) and Tcast2b (179-bp repeat) are tandemly arranged within pericentromeric regions, whereas Tcast2c (300-bp repeat) is preferentially dispersed within euchromatic regions. TCAST2 subfamilies show either overrepresentation or almost complete absence of a particular subfamily in some strains [128]. Population-specific *T. castaneum* satellites are proposed to have arisen from homologous recombination, probably stimulated by environmental stress and partial organization of TCAST2 satellite DNA in the form of single repeats dispersed within euchromatin [128,136].

#### 2.1.3. Telomeric and Subtelomeric Satellites

DNA polymerases replicate DNA from 5′-to-3′ so cannot by themselves fully replicate the very ends of chromosomes, which can cause telomere shortening at every replication round. In the large majority of eukaryotes, this end-replication problem is prevented by telomerase, which adds de novo telomeric repeats to the G-rich 3′-overhang [137]. A loss of telomere maintenance function leads to telomere shortening, which results in human diseases such as bone marrow failure, cancer and premature aging. In humans, telomerase is active only in germline and in stem cells but not in somatic cells, limiting the number of cell divisions to reduce cancer proliferation and the risk of accumulating mutations that could lead to genomic instability and tumorigenesis. Most telomeric repeats fall into the category of microsatellites, whereas subtelomeric repeats are categorized as satellites. Telomeres consist of short tandem repeats of 5′-TTAGGG-3′ in most vertebrates, 5′-TTAGG-3′ in most insects and 5′-TTTAGGG-3′ in plants [11,138,139,140]. Telomeres in humans consist of thousands of repeats of 5′-TTAGGG-3′ ending in a G-rich 3′-overhang 30–300 nucleotides in length [141]. Telomeric motifs are also present within chromosomes, where they are called interstitial telomeric sequences. In contrast to telomeres from most groups of organisms, *Drosophila* telomeres are maintained without encoding telomerase, but rather have domesticated retrotransposons for this purpose [142]. *Drosophila* telomeres are composed of three subdomains—a cap that distinguishes telomeres from a DNA double-strand break, retrotransposon arrays that maintain telomere length by targeted transposition to telomeres, and proximal telomere-associated sequences (TAS) consisting of a mosaic of complex repeats [143] Telomeric retrotransposon arrays consist of up to 12 kb of tandem arrays of telomere-specific LINE-like elements: HeT-A, TART and TAHRE [144,145,146]. Telomeres of lower Diptera, *Anopheles*, *Rhynchosciara,* and *Chironomus* are different from both microsatellite telomeres of most eukaryotes and retrotransposon telomeres of *Drosophila*. Lower Diptera telomeres carry long tandem repeats (satellites) and telomere elongation and maintenance has been suggested to take place by unequal crossover during homologous recombination [147,148,149,150].

Subtelomeric sequences vary among eukaryotes and between different chromosomes within a species. Comparative analysis of fully sequenced subtelomeres of human 4p, 16p, 22q, Xq and Yq, has revealed the presence of proximal and distal subtelomeric domains that are separated by a stretch of degenerate TTAGGG repeats [151,152,153]. While the proximal TAS subtelomeric domain is gene-poor and shows much longer uninterrupted homology to a few chromosome ends, the telomere-distal subtelomeric domain is found at only a few chromosome ends and contains nonessential genes [151,154,155,156]. Additionally, human subtelomeric domains also show high copy number variation [157].

### 2.2. Roles of Satellite DNA Sequences

The exact role of most satellite DNA sequences remains unclear in the processes in which they are involved because of their rapid evolution and technical intractability. However, a few protein-binding sequence motifs have been identified in satellite DNA. Another DNA sequence feature is the potential to fold into non-B-form DNA secondary structures, and some satellites are enriched for predicted DNA secondary structures.

#### 2.2.1. DNA Sequence Motifs

Thus far, two protein-binding satellite sequence motifs, CENP-B-box and pJα direct repeat, have been described for α-satellites in primates. The mammalian CENP-B box is a 17-bp sequence motif (5′-CTTCGTTGGAAACGGGA-3′) that is bound by the conserved CENP-B protein both in vitro and in vivo [158,159,160]. CENP-B protein was evidently domesticated from a pogo-like transposase, which was proposed to have been a source of ancestral genetic variation among mammalian satellites [51,161,162]. Another sequence motif, a 9-bp alphoid-derived direct repeat (GTGAAAAAG), is found at the junction of α-satellite and pericentric satellites in chromosomes 13, 14, and 21 and binds to a putative α-satellite DNA binding protein, pJα [163,164]. Whether pJα has a role in centromere function is unknown.

#### 2.2.2. DNA Secondary Structures

Predicted DNA secondary structures are highly enriched on centromeric satellites of the mouse and primates (Figure 1) [165,166]. Human centromeric α-satellites and the C-rich strand of *D. melanogaster* dodeca satellite are predicted to form dimeric i-motif structures generated by the association of two parallel DNA duplexes combined in an antiparallel manner in vitro [166,167]. Inverted and palindromic repeats and especially dyad and cruciform structures can potentially act as nucleosome-positioning signals as an alternative to the intrinsic curvature of the DNA and such structures have been predicted in satellites of several insects, the parasitic wasps (*Diadromus* and *Eupelmus*), *Tribolium* sp., *Trichogramma brassicae* and *Chironomus pallidivittatus*) [168,169,170,171,172]. The predicted DNA curvature resulting from A-T tracts is a strong feature of the pericentric satellites in several insect species (Figure 1) [125,168,173,174]. The potential role of predicted DNA secondary structures and curvature within satellite DNA is not yet established, but such regions may serve as the binding surface for DNA-bending proteins such as CENP-B [165,175].

The G-rich strand at the 3′ ends of the chromosome at telomeres is longer than the C strand, forming a 3′ G-overhang that is bound by the hexameric shelterin complex [176]. Shelterin regulates telomerase to prevent telomere loss and inhibits DNA repair to avoid telomere ends being mistaken for damaged DNA and activating the DNA damage response [176,177]. The G-overhang allows the formation of a tertiary structure termed the t-loop, where the G-overhang folds back and inserts into the upstream duplex telomeric DNA, displacing the G-rich strand, thereby hiding the end of the chromosome (Figure 1) [178,179,180]. According to the t-loop end protection model, the shelterin component TRF2 promotes end protection by binding and stabilizing t-loops [181].

Telomeric short tandem repeats of the G-rich strand can form into secondary structures in vitro, such as a four-stranded G-quadruplex structure derived by intramolecular folding of the G-rich single-stranded extension (Figure 1). G-quadruplex structures have been identified in vivo in ciliates but the direct evidence of G-quadruplex structures in mammalian cells is still lacking [182,183]. Indirect support for the existence of G-quadruplexes at human telomeres has come from experiments with G-quadruplex stabilizing ligands that result in the depletion of telomeric proteins, including shelterin complex components [184,185,186,187]. Telomeric G-quadruplexes have been speculated to play a role in processes that include telomere associations, such as the sister chromatid cohesion during DNA replication or telomere clustering into a “bouquet” during meiosis [188,189,190].

## 3. Satellite Chromatin

Satellite DNA biology has remained unclear due to the technical intractability of long rapidly evolving tandem repeat arrays, so that the focus of functional studies has largely shifted to satellite chromatin. Based on structure and function, chromatin has been cytologically classified into less compact euchromatin and more compact heterochromatin. Satellite regions are primarily heterochromatic in nature but can be transcribed into non-coding RNA (ncRNA). The nucleosomes of satellite heterochromatin are associated with histone variants, DNA methylation, and histone modifications that prevent transcription. The structure of centromeric chromatin, which marks centromeres, is distinct from both euchromatin and heterochromatin. Heterochromatin and centromeric chromatin are bound by distinct protein components that contribute to the establishment of underlying chromatin states (Figure 2) [191,192,193].

### 3.1. Centromeric Chromatin

Although centromeric DNA sequences are not evolutionarily conserved, most eukaryotic lineages assemble nucleosomes in which histone H3 is replaced by its (cenH3) variant, Centromeric protein A (CENP-A) [194,195,196]. CENP-A nucleosomes serve as the chromatin foundation for the kinetochore, an assemblage of multi-protein complexes that make direct connections with growing and shrinking spindle microtubules to mediate chromosome segregation [197,198]. CENP-A is loaded onto centromeres by CENP-A specific chaperone Holiday Junction Recognition Protein (HJURP); however, it is not known how HJURP recognizes centromeric satellite sequences. HJURP was originally described as a protein that binds to cruciform structures representing recombination intermediates, and we have proposed that HJURP recognizes centromeric satellites through their predicted cruciform DNA structures that are enriched on satellite centromeres and neocentromeres [165].

Only a subset of α-satellite arrays on a given chromosome assemble CENP-A chromatin and are functional [49,199]. The majority of CENP-A nucleosomes are assembled on abundant SF1 dimeric α-satellite arrays, where they are precisely and strongly positioned [49]. As these dimeric arrays contain the densest CENP-B boxes, they are expected to bind to high amounts of CENP-B. Indeed, SF1 dimers are associated with high levels of CENP-B that are correlated with high levels of CENP-A assembled on them, suggesting that CENP-B binding enhances CENP-A recruitment [49,196,200]. However, the presence of a CENP-B box is not necessary for the formation of CENP-A chromatin as Y centromeres and neocentromeres that lack CENP-B boxes and CENP-B protein efficiently assemble centromere chromatin [201]. SF2 HOR α-satellite arrays assemble intermediate levels of CENP-A [49]. Centromeres containing these HORs are fully functional, suggesting that CENP-A amounts present on them are sufficient for proper centromere function [199]. Indeed, a single human centromere is shown to contain only ~400 CENP-A molecules [202]. Monomeric α-satellites that are present at the centromere edges proximal to pericentric regions contain very low levels of CENP-A, possibly arising from the residual spreading from the core centromeres [49,200,203].

CENP-A nucleosomes are tightly associated with other DNA-bound centromere-specific proteins CENP-B, CENP-C and the CENP-TWSX histone-fold complex to form a coherent assemblage (Figure 2) [203,204,205], resulting in protection of more than nucleosomal length DNA [200]. CENP-A chromatin is efficiently recovered under high salt conditions, although a small fraction of CENP-A is found in transiently incorporating intermediate particles that are sensitive to high salt [200,206]. Centromeric satellites are transcribed into ncRNAs that are necessary for centromere function chromosomes segregation [207,208,209]. Centromeric transcripts are involved in the structural stabilization of the centromeric chromatin [206,208,210]. How centromeric chromatin might affect the transcription of underlying satellites is not yet understood.

The patterns of centromeric DNA methylation are variable across different species and tissues. In mice, DNA methylation levels vary depending on tissue type with somatic cells, sperm and oocytes containing high, intermediate and low DNA methylation levels, respectively [211]. Human and *Arabidopsis* centromeric satellites are significantly methylated, but rice and maize centromeric satellites are hypomethylated [212,213,214,215]. A recent study showed that the Japanese killifish medaka (*Oryzias latipes*) contains a specific pattern of hypomethylated and hypermethylated domains in centromeric repeats [216]. Human CENP-B preferentially binds to the unmethylated CENP-B box DNA rather than the methylated form. DNA methylation of the CENP-B box reduces human CENP-B binding and the DNMT inhibitor, 5-aza-2′-deoxycytidine, leads to the redistribution of CENP-B [217]. However, CENP-C, another component of the CENP-A chromatin complex, interacts with the de novo DNA methyltransferase DNMT3B, and both CENP-C and DNMT3B are involved in repressing centromeric transcription [218]. Further work is needed to determine whether or not repression of transcription by DNA methylation in these cases helps to maintain functional centromeres.

### 3.2. Pericentric Heterochromatin

When protein-coding genes are juxtaposed to pericentric heterochromatin due to genomic rearrangements, they undergo silencing in some somatic cells but not in others, a phenomenon called position-effect variegation (PEV) [219,220,221]. Human pericentric satellites are silenced in normal cells but are aberrantly expressed in many cancer cells, which coincides with the accumulation of methyl-CpG binding protein 2 at pericentromeres [222,223]. Pericentromeric satellites assemble highly condensed constitutive heterochromatin, which once established during development, remains permanently condensed during interphase. In contrast, facultative heterochromatin is assembled on developmentally regulated loci, which can switch between compacted and decompacted chromatin states during gene repression and activation. Pericentric heterochromatin serves to maintain sister chromatid cohesion around centromeres to prevent premature chromatid separation during chromosome segregation [224]. Pericentric heterochromatin is characterized by methylation of Lysine-9 of histone H3 (H3K9me) and trimethylation of Lysine-20 of histone H4 (H4K20me3) catalyzed by Suv29H1 and SUV4-20H1&2 histone methyltransferases, respectively (Figure 2) [225,226]. Pericentric chromocenters can be seen as large round structures upon H3K9me staining [116,227]. The presence of H3K9me3 on pericentric heterochromatin provides binding sites for heterochromatin protein 1 (HP1), which then recruits SUV4-20H which trimethylates H4K20 [225,228,229]. Pericentric satellites are uniformly DNA hypermethylated across different species and tissues, making pericentric chromatin environment repressive [211].

Despite its highly condensed structure, pericentric heterochromatin is transcribed into non-coding RNAs. Mouse MaSats are transcribed from both strands into ncRNAs of heterogeneous length [230]. During embryogenesis, pericentric satellites undergo a transient peak of strand-specific transcription precisely at the time of chromocenter formation during the 2-cell stage. Interference with MaSat transcripts using locked nucleic acid (LNA)-DNA gapmers results in developmental arrest before completion of chromocenter formation [231]. Mouse pericentric satellites are also expressed during spermatogenesis to produce MaSat repeat transcripts that are involved in the recruitment of SUV39H2, which trimethylates H3K9 on the pericentric heterochromatin required for meiotic chromosome segregation [232]. Additionally, transcription from MaSats increases significantly during neuronal differentiation, implying that satellite transcripts might play a role in differentiating post-mitotic neurons [233]. MaSat transcripts remain associated with chromatin and have a secondary structure that favors RNA:DNA hybrid formation [234]. MaSat transcripts participate in the recruitment to scaffold attachment factor B (SAFB) to pericentric regions [235]. H3K9me3 chromatin signals on pericentric satellite decrease in the absence of structural RNA [236,237,238]. Together these results suggest that pericentric heterochromatin is permissive to transcription, and the resulting transcripts may act as a structural scaffold to maintain the native condensed state of heterochromatin.

Apart from their function in sister chromatin cohesion, pericentromeric regions are also proposed to be involved in cellular differentiation or reprogramming. Mouse MaSats and human γ-satellites on chromosome 8 contain multiple recognition sites for the zinc finger protein, Ikaros [239]. Ikaros binds directly to both the promoters of many lymphoid-specific genes and pericentromeric satellite DNA in actively dividing mouse hematopoietic cells [239,240,241,242]. These results have led to the model that Ikaros facilitates repression of many lymphoid-specific genes by targeting them to pericentric heterochromatin [239,240,241,242]. Interestingly human pericentric γ-satellites that carry CTCF and Ikaros binding sites allow transcriptionally permissive chromatin in the nearby transgene to protect it from heterochromatic silencing [243].

### 3.3. Telomeric and Subtelomeric Heterochromatin

Similar to pericentric regions, reporter genes introduced in proximity to telomeres undergo silencing, suggesting that the telomeric chromatin environment is repressive [244,245]. Telomeric satellites assemble an unusual chromatin structure characterized by irregularly spaced and tightly packed nucleosomes separated by 10–20 bp DNA linkers (Figure 2) [246,247,248]. In most but not all eukaryotes, telomeric repeats (5–8 bp) are unfavorable for nucleosome formation, being out of phase with the 10-bp DNA double-helical periodicity that accommodates regular arginine minor groove contacts around the histone octamer surface [249,250]. Telomeric and subtelomeric regions are characterized by repressive heterochromatic marks H3K9me3, H3K79me2, H4K20me3 and HP1 binding (Figure 2) [251,252]. The telomere capping complex called the shelterin complex is tightly bound to telomeric repeat DNA and contributes to protecting of telomeric ends by compacting telomeric chromatin [253]. In *Drosophila* HP1, encoded by Su(var)205, is part of the telomere capping complex, called terminin, which includes rapidly-evolving telomere-specific proteins that prevent telomere fusion [254]. HP1 is recruited to telomeres independently of the presence of H3K9me3 and spreads into neighboring retroelements arrays where HP1 chromodomain binds H3K9me3 [255,256]. The loss of heterochromatic marks at telomeres and subtelomeric regions are linked to elongated telomeres [251,257]. Additionally, Retinoblastoma (RB) family proteins are also involved in promoting H4K20me3 and DNA methylation levels at both telomeres and pericentromeres by promoting the recruitment of Suv4-20h HMTase and HP1 to these loci [258]. Lack of the Rb family proteins results in decreased H4K20 tri-methylation, which coincides with chromosome segregation defects and telomere elongation [258,259]. Telomeric sequences lack CG dinucleotides, the known substrates for mammalian DNMTs, and therefore telomeres lack DNA methylation. However, subtelomeric sequences are heavily methylated and are proposed to contribute to telomeric position-effect silencing and regulation of homologous recombination and telomere length [245,260,261,262]. A decrease in subtelomeric DNA methylation leads to extreme telomere elongation [261,262]. Although deficiency of DNMTs does not alter the levels of histone methylation at telomeres, it causes a dramatic increase in telomere recombination and telomere length [262].

Telomeric and subtelomeric sequences in vertebrates are transcribed by RNA polymerase II into UUAGGG-repeat-containing RNAs (TERRAs or TelRNA), which act as structural and regulatory components of telomeres [263,264]. TERRAs are heterogeneous in length (100–9000 bp) and localize to telomeric chromatin in cis, a feature reported earlier for the XIST ncRNA, which mediates mammalian X-chromosome inactivation [264,265]. The TERRA levels increase when HMTases Suv39h and Suv4-20h are depleted, suggesting that the heterochromatin silences telomeric repeats [263,266]. Telomeric RNAs have been proposed to control telomere structure as well as telomere elongation by telomerase [264,267]. Interestingly, G-rich human TERRA form G-quadruplexes that might inhibit telomerase activity, similar to telomeric DNA [268,269,270,271].

## 4. Satellite Evolution

Satellite DNA sequences undergo high evolutionary turnover involving both stochastic events and selective pressures [30,52]. The high rate of satellite evolution produces species-specific satellite families that can be used for species identification and inferring phylogeny [19,191,261,272,273,274,275]. Several molecular mechanisms underlie satellite sequence complexity, homogeneity, array size variability, and rapid evolution (Figure 3).

### 4.1. Library Model for Satellite DNA Evolution

According to the library hypothesis of satellite DNA evolution, closely related species share a set of conserved satellite DNA families that originated in the ancestor, but each of which is differentially amplified in each species [276,277]. Amplification of a given satellite DNA family in one species will make it appear as a low-copy satellite in sister species. A simple amplification event will allow higher monomers in the related species to remain significantly conserved. Interspecific sequence conservation and absence of species-specific mutations was first demonstrated in four *Palorus* congeneric species separated by long evolutionary periods of up to 60 million years [276]. Each of these *Palorus* species contains a single pericentromeric AT-rich satellite DNA on all chromosomes, accounting for 20–40% of the genome. All four sequences are present in each of these *Palorus* species and show high conservation of sequences, repeat length and organization. In a given *Palorus* species, one of the four satellites is amplified while the three others are present as low-copy-number repeats accounting for ~0.05% of the genome. The low-copy-number satellites are dispersed between the large arrays of the major satellite over the whole heterochromatic block. The library model has since been confirmed in plants, mammals, nematodes and insects [173,278,279,280]. Satellite generation and amplification are proposed to result from the activity of transposable elements and replication of extrachromosomal tandem repeat circles by rolling-circle replication and reinsertion into the genome by unequal crossing over [281,282,283,284,285].

### 4.2. Concerted Evolution of Satellites

Given that satellite DNAs are among the fastest evolving DNA, different species acquire species-specific satellite families. The amount of satellite DNA as a fraction of the total genome size also varies drastically among closely related species. For example, *Drosophila simulans* has 5% of the genome composed of satellite DNA while only the 0.5% of the *D. erecta* genome is satellite DNA [286]. However, certain satellite families are shared by several species in a genus e.g., α-satellites in primates. In the absence of selective forces, once a given sequence has been amplified into multiple copies in the genome, each copy can incorporate mutations independently and become diverged from each other, leading to a high divergence within a species. However, members of a satellite family show a low divergence rate because satellite repeats undergo concerted evolution in which monomers of a satellite family are homogenized within a species. The outcome of this homogenization is either identical tandem repeats as in mouse telomeres or dimers or higher-order repeats as in human centromeric regions. A few groups of human chromosomes are undergoing concerted evolution as they share large arrays of satellites (1, 5 & 19, 13 & 21, 14 & 22) [287,288].

Several models have been proposed to explain the evolution of satellite DNAs. Crow and Kimura proposed a population model assuming that stabilizing selection would operate to maintain copy numbers [289]. According to their model, the distribution of repeat numbers per genome should reach an equilibrium point stabilized by unequal crossing-over and stabilizing selection, while the selection for optimal array size could be the major player of the evolutionary outcome [289]. A frequently cited model for satellite organization and evolution was developed by Smith using computer-generated simulations [290,291]. Smith argued that DNA sequences that are not constrained by selection have the natural tendency to become repetitive through unequal crossover processes. Smith simulated the effects of repeated out-of-register crossover between sister chromatids over an initially nonrepetitive sequence and showed that the satellite periodicity was the inevitable outcome. The Smith model was further elaborated by Perelson and Bell by including the effects of a variable mutation rate relative to the fixation time of a repeat variant [292]. They found that repeating units become heterogeneous if the mutation rate exceeds the rate of spread of a variant through an array. However, if the fixation rate exceeds the mutation rate, the resulting repeat arrays are homogenous due to the higher likelihood of stochastic removal of mutations before amplification [292]. The Smith model was further elaborated by Stephan (1989) and Stephan and Cho (1994), who showed that for a given sequence length, the out-of-register recombination to generate long repeat arrays is allowed if the mitotic recombination rate between sister chromatids is higher than the base pair mutation rate [293,294].

Dover and colleagues argued that concerted evolution occurs via molecular drive, an evolutionary process emerging from the activities of a number of ubiquitous mechanisms of DNA turnover, such as gene conversion, unequal crossing over, replication slippage, rolling circle replication, and multiple TE insertions [295,296,297]. A specific molecular drive mechanism based on known properties of centromeric chromatin was recently proposed by Rice, who argued that collapse of the replication fork when it encounters CENP-A nucleosome-associated complexes and/or tightly bound CENP-B results in DNA breaks that are repaired by out-of-register reinitiations [298]. Such break-induced recombination (BIR) had been described for expansions and contractions of rDNA arrays, induced by fork collisions with RNA Polymerases [299]. At centromeres, BIR may also lead to ratcheting up of copy number as CENP-A complexes are assembled onto newly replicated DNA, increasing the probability of further collisions, breaks and reinitiations. It is also possible that both BIR and unequal reciprocal recombination contribute to satellite evolution, for example, expansions by BIR and deletions by unequal exchange.

### 4.3. Centromere Drive

Whereas molecular drive might account for satellite DNA evolution under conditions of relaxed selection, centromeres are under intense selection, in that loss of a single centromere or of a centromere protein results in death of the cell. The centromere drive model was introduced to explain the rapid evolution of both centromeric DNA sequences and essential centromere proteins [29,300]. During female meiosis, only one of the four gametic products will be included in the egg, whereas the other three products will be lost in the polar body. This process of meiotic drive was originally envisioned to explain selfish non-Mendelian segregation of heterochromatic maize knobs [301], and was applied to centromeres by proposing competition for the egg pole by centromere-directed reorientation of the meiosis I tetrad. For example, variants of centromeric satellite arrays that enhance the spindle binding during chromosome segregation in meiosis have been observed to selfishly drive their own transmission in this fashion [302,303,304]. It is also possible that specific satellite sequence variants that increase the efficiency of proper spindle attachment increase their frequency in the population. For example, human centromeric dimeric arrays might be expanding due to their more efficient recruitment of centromeric protein shown by centromeric chromatin profiling [49,196,200,203]. Analogous to centromeric satellites, subtelomeric satellites of different organisms show no sequence similarity and have been proposed to be subjected to telomere drive in which rapid sequence changes in the subtelomeric DNA can trigger adaptation of telomeric proteins to restore telomere homeostasis [305,306,307].

### 4.4. Repeat Evolution and Speciation

Bateson–Dobzhansky–Muller incompatibilities between rapidly evolving gene pairs contribute to the hybrid failure (sterility/inviability) and speciation [308]. For example, *D. simulans* Lethal hybrid rescue (Lhr) and *D. melanogaster* Hybrid male rescue (Hmr) have functionally diverged and interact with each other to cause lethality in F1 hybrid males [309]. LHR is adaptively evolving and localizes to pericentric regions, suggesting that rapidly evolving heterochromatic satellite DNA sequences may be driving the evolution of the incompatibility genes [309,310]. Hmr and Lhr encode proteins that form a heterochromatic complex with HP1and repress transcripts from satellite DNAs and many families of transposable elements [310]. Hmr and Lhr mutations also cause massive overexpression of telomeric TEs and significant telomere lengthening. Hmr and Lhr regulate three major types of heterochromatic sequences responsible for the significant differences between *D. melanogaster* and *D. simulans* and thus have high potential to cause genetic conflicts with host fitness. Adaptive divergence of heterochromatin proteins in response to satellite DNA divergence might be an important underlying force driving the evolution of hybrid incompatibility genes. Direct evidence for the involvement of satellite DNA in speciation came from study where *D. melanogaster* locus Zhr, which maps to the pericentric 359-bp satellite array on the X chromosomes, caused lethality in F1 daughters from crosses between *D. simulans* females and *D. melanogaster* males [311]. Hybrid females show mitotic defects due to lagging X chromatids during early embryogenesis. A deletion of the entire 359-bp satellite array leads to the normal segregation of X chromosomes, while translocation of the 359-bp satellite array to the Y chromosome results in Y chromosome segregation defects. The 359-bp satellite array is absent in *D. simulans,* suggesting that the absence or divergence of factors for required for species-specific heterochromatin formation in the *D. simulans* maternal cytoplasm causing F1 female lethality.

## 5. Conclusions and Future Perspective

Satellite DNA repeats are among the most diverse and versatile genomic sequence classes contributing to several key biological processes, and satellite deregulation is seen in various cancers and other diseases. Satellites are hypertranscribed in lung, kidney, ovary, colon, and prostate cancers, possibly due to loss of silencing and hypomethylation of satellite DNA being associated with Wilms tumors, mammary adenocarcinomas, ovarian epithelial carcinomas and immunodeficiency-centromeric instability-facial anomalies (ICF) syndrome [312,313,314,315,316]. However, satellites remain less studied than non-repetitive regions because their highly repetitive nature makes their genetic manipulation challenging and rapid satellite evolution makes identification of conserved sequence motifs difficult. Furthermore, linear assembly maps for satellite regions and the knowledge of the full satellite complement of insect, mammalian and plant species remains incomplete. However, with the improvement of long-read sequencing technology, it has recently become possible to obtain end-to-end maps of the human X chromosome that span centromeric and telomeric regions [317]. Additionally, using the long-read sequencing, the centromeric satellites have been assembled for chromosome 8 and utilized for evolutionary analysis of α-satellite arrays [318]. The rapid improvement in long-read sequencing technology should soon reveal the precise arrangement of satellite blocks, which will provide evidence of the rearrangements that have occurred during their emergence and evolution. Transcription of satellite chromatin, whether heterochromatin or specialized telomeric chromatin, is poorly understood relative to transcribed euchromatin, but advances in understanding gene transcriptional regulation should be applicable to satellites. With improving chromatin extraction and genomic profiling technologies, it is now possible to determine the conformation of satellite chromatin complexes at base-pair resolution [200,203]. We also look forward to better understanding of satellite recognition and determining the degree to which DNA secondary structures are involved in satellite biology at both centromeres and telomeres.

## Figures and Tables

**Figure 1 ijms-22-04309-f001:**
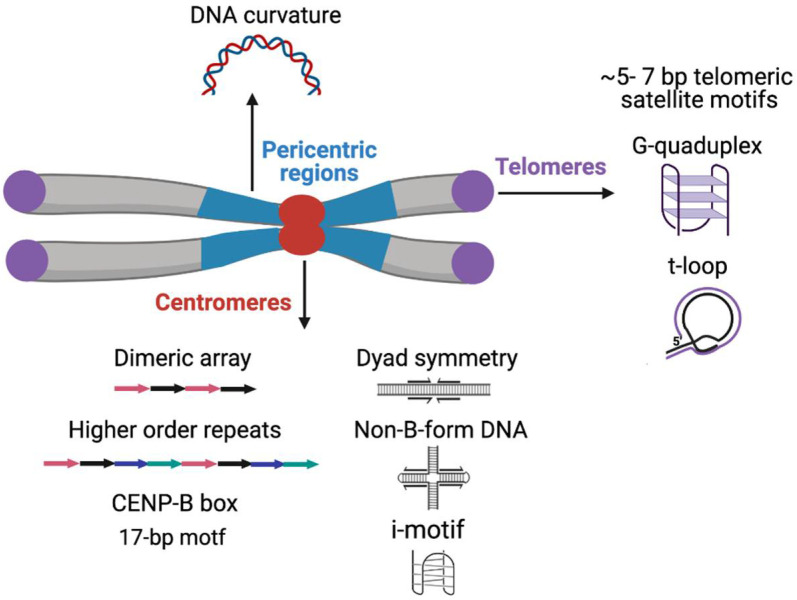
Sequence motifs, basic organizational units and predicted non-B-form secondary DNA structures on various functional classes of satellite DNA.

**Figure 2 ijms-22-04309-f002:**
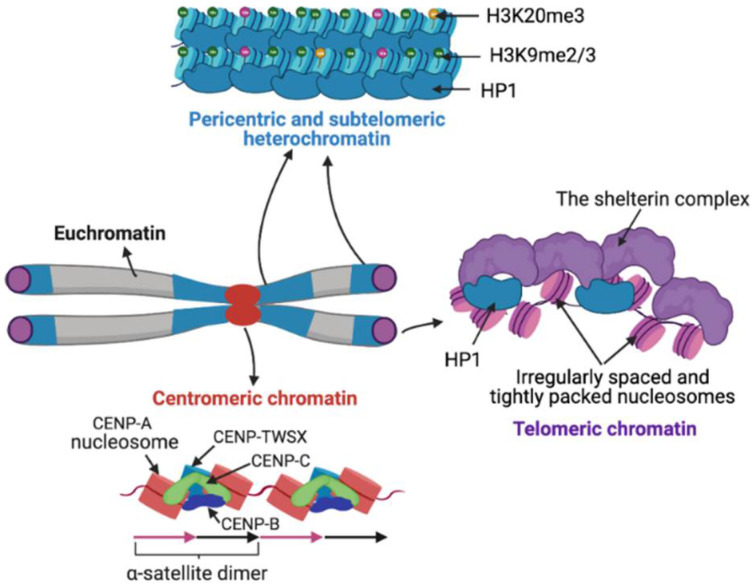
Satellite chromatin organization. In contrast to euchromatin, pericentric and subtelomeric regions are compact, marked by repressive modifications such as H3K9me3 and H3K20me3 and bound by Heterochromatin-associated Protein 1 (HP1). At centromeres, Centromere Protein A (CENP-A) nucleosomes form tight complexes with DNA binding centromeric proteins CENP-B, CENP-C and the histone-fold containing protein complex CENP-TWSX. Telomeric ends contain irregularly spaced and tightly packed nucleosomes separated by 10-20 bp DNA linkers. Telomeric chromatin is tightly associated with the shelterin complex and HP1.

**Figure 3 ijms-22-04309-f003:**
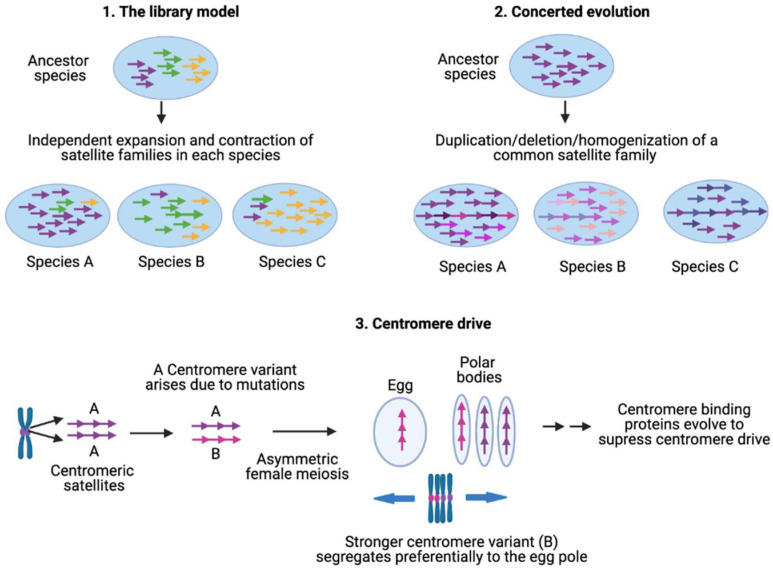
Satellite evolution. The library model can account for the divergent evolution of species-specific satellite families through independent expansion and contraction of satellite families in diverging species. Concerted evolution occurs via recombinational processes that result in expansions, contractions and homogenizations of tandemly repetitive sequences. Centromere drive is a selfish process in which stronger centromeres preferentially orient towards the egg pole during asymmetric female meiosis. Drive is suppressed when mutation of a centromere-binding protein restores centromere balance, resulting in an arms race between centromeric satellite DNA and centromere proteins.

**Table 1 ijms-22-04309-t001:** Centromeric satellite repeat units of mammals, plants and insects.

Organism	Repeat Unit	Length	References
Human	α-satellite	171-bp	[4]
*M. musculus*	Minor satellite	120-bp	[34,35]
	MS3	150-bp	
	MS4	300-bp	
*Z. mays*	CentC	156-bp	[36,37,38]
	CRM	~7000-bp	
*O. sativa*	RCS2 or CentO	155-bp	[39]
	CRR	4400- and 9200-bp	
*L. nivea*	LCS1	178-bp	[40]
*A. thaliana*	pAL1	178-bp	[41,42]
*R. sativus*	pRA5/pRB	~177-bp	[43]
*D. melanogaster*	Prodsat 10-mer AATAACATAG	10-bp	[44,45]
	AATAT	5-bp	
	AATAG	5-bp	
	AAGAT	5-bp	
	G2/Jockey-3	Up to a few kb	
*D. simulans*	10-mer AATAGAATTG	10-bp	[44,45]
	Simcent1	154-bp	
	Simcent2	88-bp	
	G2/Jockey-3	Up to a few kb	

**Table 2 ijms-22-04309-t002:** Pericentromeric satellite repeat units of mammals, plants and insects.

Organism	Repeat Unit	Monomer Length	Reference
Human	Satellite I	17-bp, 25-bp	[23,62,102,105]
	Satellite II	10–80-bp	
	Satellite III	5-bp, 10-bp	
	α Satellites	68-bp	
	α Satellites	220-bp	
*M. musculus*	Major satellites	234-bp	[24,34]
	TRPC-21A	21-bp	
	TLC	146-bp	
*A.thaliana*	pAL1/pAtMr	180-bp	[42,117,118,119]
	Athila retrotransposons	Up to 14,100-bp	
	500 bp repeat	500-bp	
	160 bp repeat	160-bp	
	Dispersed repeats-163A, 164A, 278A and 106B	1000–2000-bp	
*S. bulbocastanum*	2D8	5900-bp	[120]
*D. melanogaster*	1.688 gm/cm^3^ buoyant density family	353-bp, 356-bp, 260-bp and 359-bp	[121,122,123]
	Rsp	120-bp + 120-bp	
	Prod	10-bp	
	AAGAT	5-bp	
	AATAG	5-bp	
	AAGAG	5-bp	
*P. subdepressus*	T2A5T octanucleotide containing 72-bp-long repeat	8-bp, 22-bp	[124]
*C. americana*	189 bp repeat	189-bp	[125]
*C. carnifex*	Complex satellite repeat	211-bp, 477-bp, 633-bp	[126]
*T. castaneum*	TCAST1	377-bp and 362-bp	[127,128]
	Tcast2a	359-bp	
	Tcast2b	179-bp	
	Tcast2c	300-bp	

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
