# Peer review of "Sequence, Chromatin and Evolution of Satellite DNA"

_ijms, 2021, doi:10.3390/ijms22094309_

Round 1

Reviewer 1 Report

In their review article, Thakur, Packiaraj and Henikoff summarize the current understanding of satellite DNA sequences and satellite DNA-associated chromatin, providing also a summary of satellite DNA evolution concepts.

The review is clearly structured and correctly written. Although I find some sections of the review too example-listing with very little discussion of the possible avenues for future research, I do not have major concerns.

I have several comments that are listed below in order of appearance in the MS:

Line 49-53

Given that the authors elaborate CsCl gradient and reassociation kinetics methods in some detail, I would suggest to mention also some of the bioinformatics tools (e.g. Tandem Repeat Finder, RepeatExplorer, TAREAN…) that have revolutionized satellite DNA research in the last decade.

Line 153-174 (section 2.1.1.3. Plant centromere satellites),

including also line 254-270 (section 2.1.2.2. Plant pericentric satellites)

Although the review cites a lot of papers, I would prefer to see more recent citations, especially from the field of plant satellite DNA biology, which is very propulsive at the moment.

Line 175-177

The authors state that “among nearly 1 million known insect species, satellite DNA has been studied for only a few, and centromeric satellites have been studied only in D melanogaster and its sister species, D. simulans”.

I do not agree with any of these statements. The satellite DNAs have been studied in many insect species, and the two Drosophila species are not the only insects whose centromeric satellite DNAs have been studied so far. The authors might want to cite some other examples too.

Line 314-344

Speaking about telomeric and subtelomeric satellites, the authors could comment the case of the mosquito Anopheles gambiae and its satellite-based telomeres.

Line 405-406

Certain references should be added (e.g. Blower et al (2002) Dev Cell. 2:319, Sullivan & Karpen (2004) Nat Struct Mol Biol 11:1076…).

Line 565-568

In addition to the cited reference, Fry & Salser (1977) Cell12:1069 citation should be added as it was the first paper introducing the satellite DNA library model.

MINOR ISSUES:

Line 20-22 (Abstract)

Summarizing their work, the authors state: “We also discuss how the evolution of functional satellite DNA classes may contribute to speciation in insects, plants and animals.”

As the insects are animals as well, the authors should rephrase the sentence.

Line 212

The missing italics.

Lines 220, 224, 503, 508

The incorrect writing of the Greek letters beta and gamma.

Line 716-1370

Many typos in the References section.

Author Response

Reviewer 1

In their review article, Thakur, Packiaraj and Henikoff summarize the current understanding of satellite DNA sequences and satellite DNA-associated chromatin, providing also a summary of satellite DNA evolution concepts. The review is clearly structured and correctly written. Although I find some sections of the review too example-listing with very little discussion of the possible avenues for future research, I do not have major concerns. I have several comments that are listed below in order of appearance in the MS:

Line 49-53

Given that the authors elaborate CsCl gradient and reassociation kinetics methods in some detail, I would suggest to mention also some of the bioinformatics tools (e.g. Tandem Repeat Finder, RepeatExplorer, TAREAN…) that have revolutionized satellite DNA research in the last decade.

Authors: These bioinformatics tools for Repeat analysis are now mentioned (lines 126-127).

Line 153-174 (section 2.1.1.3. Plant centromere satellites),nincluding also line 254-270 (section 2.1.2.2. Plant pericentric satellites)

Although the review cites a lot of papers, I would prefer to see more recent citations, especially from the field of plant satellite DNA biology, which is very propulsive at the moment.

Authors: We have included more recent reviews from the field of plant satellite biology (line 225).

Line 175-177

The authors state that “among nearly 1 million known insect species, satellite DNA has been studied for only a few, and centromeric satellites have been studied only in D melanogaster and its sister species, D. simulans”.

I do not agree with any of these statements. The satellite DNAs have been studied in many insect species, and the two Drosophila species are not the only insects whose centromeric satellite DNAs have been studied so far. The authors might want to cite some other examples too.

Authors: We have modified the text to reflect that satellites have been studied Diptera, Hymenoptera, Bombyx and Tribolium, and functional centromeric satellites have been functionally characterized in D. melanogaster and its sister species, D. simulans (line 241-243).

Line 314-344

Speaking about telomeric and subtelomeric satellites, the authors could comment the case of the mosquito Anopheles gambiae and its satellite-based telomeres.

Authors: Satellite telomeres of Anopheles gambiae and other Dipterans are included (lines – 384-387).

Line 405-406

Certain references should be added (e.g. Blower et al (2002) Dev Cell. 2:319, Sullivan & Karpen (2004) Nat Struct Mol Biol 11:1076…).

Authors:  References suggested by the reviewer have been added (line 454).

Line 565-568

In addition to the cited reference, Fry & Salser (1977) Cell12:1069 citation should be added as it was the first paper introducing the satellite DNA library model.

Authors:  References suggested by the reviewer have been added (line 59).

MINOR ISSUES:

Line 20-22 (Abstract)

Summarizing their work, the authors state: “We also discuss how the evolution of functional satellite DNA classes may contribute to speciation in insects, plants and animals.”

As the insects are animals as well, the authors should rephrase the sentence.

Authors: Corrected (line 67).

Line 212

The missing italics.

Authors: The original manuscript submitted by authors to IJMS contained italics. However, during the conversion of the manuscript to a different format for reviewing by the journal, italics were lost.

Lines 220, 224, 503, 508

The incorrect writing of the Greek letters beta and gamma.

Authors: The original manuscript submitted by authors to IJMS contained correct writing of the Greek letters beta and gamma. However, the conversion of the manuscript to a different format for reviewing by the journal has resulted in the loss of the Greek letters.

Line 716-1370

Many typos in the References section.

Authors: Typos in the reference section have been corrected

Reviewer 2 Report

The paper represents a comprehensive review of present knowledge related to satellite DNA sequences from different model organisms, showing their characteristics such as monomer length, primary, secondary and tertiary structure features. The second part of the paper deals with epigenetic marks linked to satellite DNA while in the third part evolution of satellite DNAs and their potential role in speciation are presented.

The review is clearly written and is very informative for the scientists who are interested in satellite DNA but also to those that are not strictly in the field. Therefore I recommend review for the publication in IJMS.

There are few minor comments that should be addressed:

  1. Lines 76-79: I think that it should be stressed that D4Z4 and other macrosatellites mentioned there belong to human. For readers not familiar with human satellites this could be valuable information.
  2. In Table 2 for Tribolium castaneum satellite TCAST1, monomer length is not written. According to cited reference no. 124 TCAST1 monomer length is 377 bp and 362 bp respectively (2 most representative TCAST1 subfamilies), and maybe this could be included in the table.
  3. At Figure 2 H3K9me3 should be written instead of H3K9me.
  4. Throughout the text the Latin version of species names should be written in italics.
  5. Line 565: “4.1. Library model for satellite DNA evolution” should be in bold as well as “4.4.  Repeat evolution  and speciation” at line 661.
  6. “3.2. Telomeric and subtelomeric heterochromatin” should be “3.3. Telomeric and subtelomeric heterochromatin”: line 511

Author Response

Reviewer 2

The paper represents a comprehensive review of present knowledge related to satellite DNA sequences from different model organisms, showing their characteristics such as monomer length, primary, secondary and tertiary structure features. The second part of the paper deals with epigenetic marks linked to satellite DNA while in the third part evolution of satellite DNAs and their potential role in speciation are presented. The review is clearly written and is very informative for the scientists who are interested in satellite DNA but also to those that are not strictly in the field. Therefore I recommend review for the publication in IJMS.

There are few minor comments that should be addressed:

Lines 76-79: I think that it should be stressed that D4Z4 and other macrosatellites mentioned there belong to human. For readers not familiar with human satellites this could be valuable information.

Authors: We have now mentioned that these macrosatellites belong to human (line 150).

In Table 2 for Tribolium castaneum satellite TCAST1, monomer length is not written. According to cited reference no. 124 TCAST1 monomer length is 377 bp and 362 bp respectively (2 most representative TCAST1 subfamilies), and maybe this could be included in the table.

Authors: We have included TCAST1 monomer length in Table 2. We thank the reviewer for pointing this out.

At Figure 2 H3K9me3 should be written instead of H3K9me.

Authors: We have changed H3K9me to H3K9me2/3 to reflect both H3K9me2 and H3K9me3 are present at the constitutive heterochromatin at pericentric and subtelomeric regions (Figure 2).

Throughout the text the Latin version of species names should be written in italics.

Authors: The original manuscript submitted by authors to IJMS contained italics. However, during the conversion of the manuscript to a different format for reviewing by the journal, italics were lost.

Line 565: “4.1. Library model for satellite DNA evolution” should be in bold as well as “4.4.  Repeat evolution  and speciation” at line 661.

Authors: The original manuscript submitted by authors to IJMS contained these headings in bold. However, during the conversion of the manuscript to a different format for reviewing by the journal, bold formatting was lost.

 “3.2. Telomeric and subtelomeric heterochromatin” should be “3.3. Telomeric and subtelomeric heterochromatin”: line 511

Authors: corrected (line 543).

Reviewer 3 Report

this is a well-written, professional and comprehensive review on an exciting topic of satellite repeats in genomes of the most important multicellular eukaryotic species, there is nothing to correct and only following formal details are to be fixed

just minor details

-the abbreviation HOR is first mentioned, line 112, and only later explained, line 114.

-italics is missing in several latin names of plants in the paragraph beginning on line 153

-a dot missing "D melanogaster", line 177

congrats on such a nice review!

Author Response

Reviewer 3

this is a well-written, professional and comprehensive review on an exciting topic of satellite repeats in genomes of the most important multicellular eukaryotic species, there is nothing to correct and only following formal details are to be fixed

just minor details

-the abbreviation HOR is first mentioned, line 112, and only later explained, line 114.

Authors: corrected (line 183-184).

-italics is missing in several latin names of plants in the paragraph beginning on line 153

Authors: The original manuscript submitted by authors to IJMS contained italics. However, during the conversion of the manuscript to a different format for reviewing by the journal, italics were lost.

-a dot missing "D melanogaster", line 177

Authors: corrected (line 246).

congrats on such a nice review!

Authors: Thank you.